# Economic and Environmental Assessment of Olive Agroforestry Practices in Northern Greece

**Emmanouil Tziolas** [1] , **Stefanos Ispikoudis** [2] , **Konstantinos Mantzanas** [3,*] , **Dimitrios Koutsoulis** [4] and **Anastasia Pantera** [2]

1   Department of Marine Fisheries, Fisheries Research Institute (FRI), Hellenic Agricultural Organization "DIMITRA", Nea Peramos, 64007 Kavala, Greece; tziolasm@inale.gr
2   Department of Forestry and Natural Environment Management, Agricultural University of Athens, 36100 Karpenissi, Greece; s.ispikoudis@aua.gr (S.I.); pantera@aua.gr (A.P.)
3   Laboratory of Rangeland Ecology, School of Forestry and Natural Environment, Aristotle University of Thessaloniki, 54124 Thessaloniki, Greece
4   Independent Researcher, 63077 Chalkidiki, Greece; dimkou1922@yahoo.gr
*   Correspondence: konman@for.auth.gr; Tel.: +30-2310-992-734

**Abstract:** Preservation and promotion of agroforestry systems entails the ideology for more ecosystem services, additional biodiversity benefits and climate change mitigation. Furthermore, farmland and forest landscapes and the consequent benefits to the environment from their combination, enhance the importance of agroforestry systems towards sustainable environmental policies. Nevertheless, traditional agroforestry systems face significant adaptation problems, especially in the EU, due to continuous economic reforms and strict agri-environmental measures. In this context our main goal is to assess the current managerial framework of two agroforestry systems and more specifically the olive agroforestry practices in Northern Greece. The economic and environmental implications of four different production plans are highlighted following the Life Cycle Costing and the Life Cycle Assessment protocols. The production plans include the simultaneous cultivation of annual crops, such as vetch and barley, along with olive groves. Potential environmental impacts are depicted in $CO_2$ equivalents, while the economic allocation of costs is divided in targeted categories (e.g., raw materials, labor, land rent, etc.). The results indicate significant deviations among the four production plans, with the combination of olive trees and barley being heavily dependent on fertilization. Furthermore, the open-spaced olive trees intercropped with a mixture of barley and commonly depicted the lowest $CO_2$ eq. emissions, though the economic cost was significantly higher than the other agroforestry system intercropped with barley only. The authors suggest that the formulation of a decision support system for agroforestry systems should be taken into account in order to preserve current agroforestry systems.

**Keywords:** agroforestry systems; olive growing systems; Life Cycle Assessment; Life Cycle Costing; Activity-Based Costing

## 1. Introduction

Agriculture represents one of the oldest and more extensive uses of human land that has supported life through time. Farming practices, developed by the timely and restless testing and observations of farmers, have defined past civilizations' fate, rendering agricultural production security as a critical factor sought for human survival [1]. Approximately two-billion people (26.7% of the world population) derive their livelihoods from agriculture [2]. Whilst increased food production is imperative, it is also quite often accompanied by negative impacts on the natural environment. Regrettably, agricultural intensification at field, farm and landscape scales, is the leading cause of deforestation resulting in shrinkage of ecosystem services, native habitat destruction, groundwater aquifer depletion and biodi-

versity reduction. The conventional agricultural model, based on crop specialization and on the massive use of external inputs, is currently facing a deep crisis [3].

Moreover, climate change is one of the biggest contemporary challenges many ecosystems face, posing a major threat to food security through its strong impact on agriculture. Agriculture is dependent on climate; therefore, farming activities will need to adapt, especially in the southern and south-eastern EU regions where the negative effects will be greater than the others [4].

Due to the adverse impacts both climate change and the conventional arable production system pose on the environment, alternative production systems are required to maintain the multifunctional landscape needed for producing food, fodder, and energy [5] (Lehmann et al. 2020). Agroforestry has been identified as one of the most promising tools capable of integrating these adverse impacts [6]. Agroforestry systems (AFS) are environmentally friendly traditional land-use practices, which combine trees and crops and/or grazing on the same land, simultaneously or sequentially. The composition of agricultural production in such systems can have diverse, mainly positive, effects in several categories (biodiversity, global warming potential, etc.) [7], therefore, alternatives must be considered, in order to develop an environmentally sustainable production plan. The inclusion of trees in agricultural areas is not a novel idea, though the necessity for agricultural sustainability, environmentally friendly practices, biodiversity conservation and carbon sequestration, have reestablished this notion as one of the most effective and sustainable interventions [8]. Agroforestry is an evolving concept used in many different environments to improve resource-use efficiency and the resilience of traditional agricultural systems, as it is supported by nature and can simultaneously provide multiple environmental, social, and economic benefits [9]. Several international bodies, for example the Global Research Alliance, FAO, IPCC (Intergovernmental Panel on Climate Change), identify agroforestry as a negative emission technology that should be expanded to reduce atmospheric GHG.

Olive is the most widespread cultivated tree in Greece, covering an area of 700.000 ha, classifying Greece as the third EU country in olive groves after Spain and Italy [10]. A great portion of 124.311 ha forms agroforestry systems with various crops or pastures established in the understory of olive trees [11]. Olive trees alone or in groves are found in all parts of the country that have a mild Mediterranean climate [12]. Intercrops of olive trees with cereals and legumes may increase the profitability and sustainability of the farm by the production of biomass and grains from the understory crops while positively affecting olive tree productivity [10,13]. This indicates intercropping combination as a promising practice that may contribute not only to increased economic returns to the farmer but also as an environmentally friendly option that decreases fertilizer use and, subsequently, soil and water contamination.

Assessing the economic and environmental sustainability of agroforestry systems (AFS) is a complicated task since the applied agricultural practices differ significantly from the relevant conventional (intensive) production. In this context, agroforestry systems are less profitable thus undermining their adoption by farmers [14] since profit maximization is the major goal for most of them [15]. On the other hand, the environmental benefits of agroforestry systems are manifold (in the form of ecosystem services, carbon sequestration, etc.) [16,17], developing an uneven regime between economic and environmental management for both policy makers and farmers.

Considering the environmental sustainability of AFS, several studies emphasize the methodological framework of Life Cycle Assessment (LCA) [18–20], underlining the applicability of LCA in AFS. Apart from the abovementioned, the holistic and analytical step-by-step procedure is suitable for characterizing agricultural activities (namely fertilization, sowing, harvesting, etc.) in one indicator expressed in $CO_2$ equivalents (GWP). Furthermore, challenges and potential improvements have been considered, studied closely and illustrated over the last years [21,22], sealing the gap of incompatibility between LCA and agricultural systems. Focusing on olive production systems in the Mediterranean basin, LCA has been implemented in order to depict the disparities of different management

practices, along with the respective impacts to the environment [5,23,24]. The emissions of agricultural activities (e.g., agrochemical application, irrigation, fertilization, etc.) are heavily dependent on the management practices of farmers, though the allocation of environmental impacts are not on par with the economic impacts [25].

Consequently, the policy framework beyond 2020 is considered inadequate and the economic lure of subsidies is not strictly connected to environmentally friendly practices on the field [26]. Thus, the economic sustainability of AFS should be investigated in addition to the magnitude of the respective impacts to the environment. The difficulty of assessing the economic impacts of AFS encompasses the intangible, in many cases, ecological and social benefits of AFS [27]. Therefore, the methodological framework for the economic assessment of AFS is usually based on a cost–benefit analysis approach [28,29], or on similar ratio approaches [27,30]. Nevertheless, the connection of LCA and Life Cycle Costing (LCC) develops a powerful tool for the holistic assessment of AFS, integrating the ideology of time and money allocation, through the life cycle of crop cultivation [31]. In this context the methodology framework should be consistent with the objectives of the study, in order to generate comparable results between LCA and LCC [32]. The main aim of the current study is the assessment of carbon footprint in conjunction with an economic analysis regarding cases of AFS in Greece, following the principles of LCA and LCC. A total of four production schemes are presented along with their respective impacts, since there is a lack of knowledge of AFS in the Greek territory.

## 2. Materials and Methods

### 2.1. Study Area

The study referred to the Kassandra peninsula of Chalkidiki (X 450,089.747 and Y 4,428,217.075) and included plots with open-spaced olive trees, monocultures of cereals and olive groves. There are also scattered agroforestry systems composed of olive trees intercropped with cereals and grasses, as cover crops, with tree densities ranging from 20 to 60 trees/ha [33]. The mean annual precipitation of the area is 602.5 mm and the mean monthly temperature is 16.2 °C [10]. The soils of the area derived from luvisols have an average pH of 8.2.

The four production systems in the Kassandra peninsula are described below:

(i) open-spaced olive trees intercropped with barley (BOT)
(ii) open-spaced olive trees intercropped with a mixture of barley and common vetch (VBOT)
(iii) olive orchards (OT)
(iv) monocultures of cereals (barley) (BF),

Olive trees of the BOT and VBOT regime were 80 years old, cultivated for the production of olives and olive oil in a density of 100 trees/ha and tree spacing of 10 × 10 m. Olive trees were pruned every year to improve olive production [10]. The crops of barley and the mixture of barley and common vetch were sowed in the autumn of three consecutive years (2014, 2015 and 2016). Seed and fertilizer quantities were the following: a. BOT and BF: barley 240 kg/ha and fertilizer 130 kg/ha (24–10–0, N–P–K) and b. VBOT: barley 80 kg/ha, common vetch 120 kg/ha and fertilizer 120 kg/ha (0–46–0, N–P–K). The barley crop was harvested early June by a harvest machine and was used as grain for livestock feeding. The mixture of barley and common vetch was cut by the end of May and harvested as hay for livestock feeding [10]. Olive orchards were 40 years old and similar to previous systems cultivated for the production of olives and olive oil. The density was 250 trees/ha and managed as VBOT and BOT production systems.

### 2.2. LCA Background

LCA is a holistic assessment and estimation framework regarding the environmental impacts throughout the lifetime of a product or process [34]. Although the International Organization for Standardization provides guidelines and principles via the published standards ISO 14040-14044 [35], the method's framework is vague especially regarding agricultural production [11]. Nevertheless, in order to conduct a complete LCA, four

interconnected phases should be introduced and defined, namely: (i) Goal and scope definition, (ii) Life Cycle Inventory (LCI), (iii) Life Cycle Impact Assessment and (iv) Interpretation [36]. The versatility of the defined methodological framework of LCA is a strong asset if used accordingly along with the integration of economic assessment via the LCC method [37]. Since the evaluation of environmental sustainability should not neglect the economic aspects of an action, especially in agriculture [38], LCC is considered as an optimal decision-making tool, when applied simultaneously with LCA.

### 2.3. Goal and Scope Definition

The main aim of the current study is a holistic environmental and economic assessment of agroforestry systems in the region of Chalkidiki, Northern Greece. The comparison among the agroforestry systems is enhanced by the depiction of single crop production plans (barley and olive trees), in order to comprehend the pros and cons of farm management in the area of interest. The framework of agroforestry is not widespread in the farming community of Greece, though such practices are considered environmentally friendly. However, maximization of income is the key element for the majority of farmers in order to adopt new or different cultivation techniques. Therefore, the gist of the study is the evaluation of trade-offs between the economic and environmental sustainability of AFS, in comparison to conventional agricultural practices, which are dominant in the area [25].

Concerning the environmental assessment, all the inputs used and their respective impacts on the field are accounted as well as the indirect impacts from raw materials extraction. Furthermore, farming activities, machinery operations and transportation of inputs and outputs to the respective destinations are also considered. The system boundaries are depicted in Figure 1, including all the relevant operations for one production (agricultural) year, starting from sowing to the harvesting procedure, including the transportation of the products to the processing unit. The cradle to factory gate variation of the system boundaries is chosen, since the expected yield in a mixed production plan could generate manifold types of outputs (e.g., kg of olives, liters of olive oil, tons of barley fodder, etc.) simultaneously. Thus, the functional unit is set to 1 ha of arable land, in order to minimize potential incompatibilities among different yield types. Apart from the abovementioned, the economic assessment follows a similar approach (EUR/ha) since area-based payments play a significant role to the subsidiary policy of the CAP [39]. In this context, four cropping management schemes will be thoroughly analyzed, comparing the emissions in equivalents, with the latest Global Warming Potential coefficients (GWP, explained below), and their economic performance in a competitive environment of progressive sustainability.

### 2.4. Life Cycle Inventory (LCI)

Formulating a credible inventory database is a complex and time-intensive procedure, while access to the appropriate datasets is considered an additional obstacle to overcome [40]. Each stage of the assessment process is conveyed to an assortment of products and services, which are expressed in units of energy per amount of output or units of emissions per area accordingly. Therefore, the necessity for accurate inventory databases regarding a variety of inputs (e.g., raw materials, energy, wastes, etc.) has motivated the generation of public and private databases [41], especially in relation to agricultural production [42]. The inventory analysis for primary energy usage and GHG emissions for the current study is presented in Table 1.

Nevertheless, the abundance of databases ranging from manifold countries to different perspectives, develops a multiplicate decision-making process in order to appropriately choose input and output units. The key elements for the formulation of a robust LCI are quality and consistency [40], taking into account credible data documentations and reliable elementary and physical flows inside the boundaries of the studied system. In relation to the reference system of the current study, the approximation of primary energy and emission outputs follows mostly the BioGrace II greenhouse gas (GHG) standard values [47], which are in line with Directive (EU) 2018/2001 [55]. The estimation of material

inputs includes fertilizers, pesticides, seeds, fuel and energy consumption for one growing season of each cropping system, while transportation of inputs and outputs, human and mechanical labor are calculated as well (Table 1). Furthermore, indirect emissions are estimated to strengthen the outputs of the study, since these are considered significant in agricultural production systems [56,57].

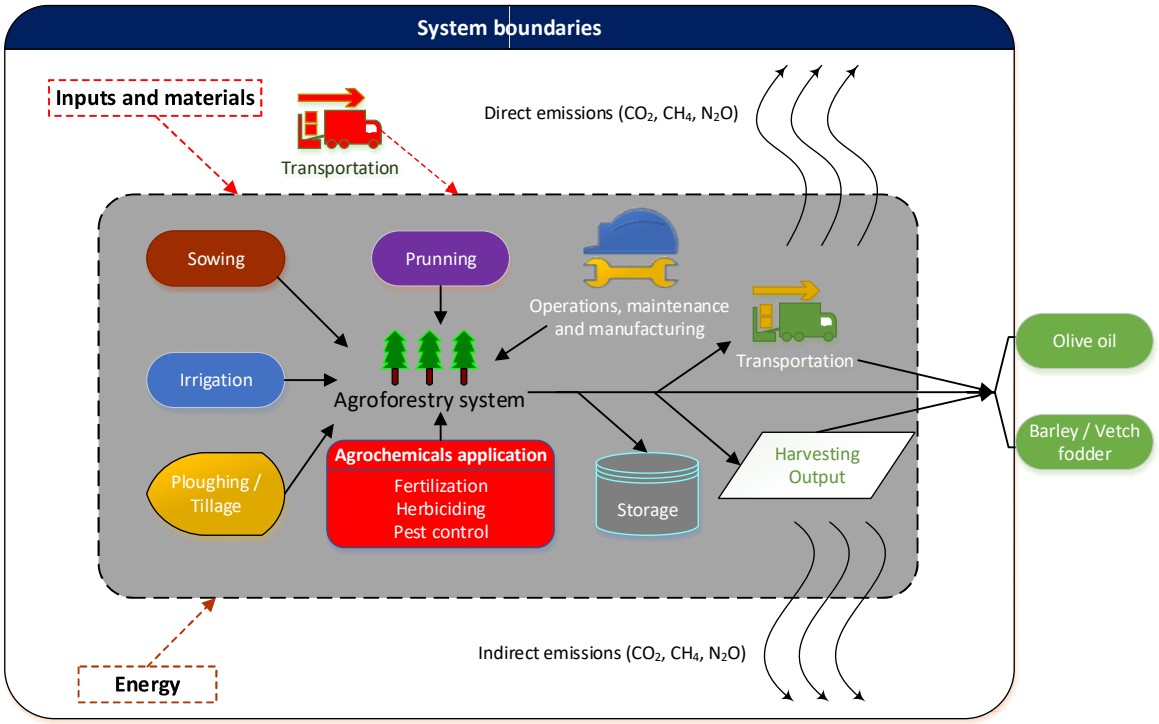

**Figure 1.** System boundaries.

**Table 1.** Inventory analysis for primary energy usage and GHG emissions.

| Inputs | Unit | Primary Energy | GHG unit | GHGs | Remarks |
|---|---|---|---|---|---|
| **Seeds** | | | | | |
| Barley seeds | MJ/kg | 2.95 | gCO$_2$eq/kg | 300.85 | [43] |
| Vetch seeds | MJ/kg | 10 | gCO$_2$eq/kg | 488.80 | [44–46] |
| **Agrochemicals** | | | | | |
| N | MJ/kg | 48.99 | gCO$_2$eq/kg | 4524.41 | [47] |
| P | MJ/kg | 15.23 | gCO$_2$eq/kg | 541,67 | [47] |
| K | MJ/kg | 9.68 | gCO$_2$eq/kg | 416.67 | [47] |
| Pesticides | MJ/kg | 268.40 | gCO$_2$eq/kg | 12,003.33 | [47] |
| **Energy** | | | | | |
| Lubricants | MJ/kg | 53.28 | gCO$_2$eq/kg | 947.00 | [47] |
| Diesel | MJ/kg | 56.80 | gCO$_2$eq/MJ | 95.10 | [47,48] |
| Petrol | MJ/kg | 60.20 | gCO$_2$eq/MJ | 93.30 | [47,48] |
| Electricity | MJ/MJ | 2.73 | gCO$_2$eq/MJ | 243.49 | [47] |
| **Operations, maintenance and manufacturing** | | | | | |
| Tractor | MJ/h | 16.42 | gCO$_2$eq/h | 9800 | [49,50] |
| Human | MJ/h | 1.80 | gCO$_2$eq | - | [49] |
| Machinery | MJ/h | 11.90–35.05 | gCO$_2$eq/h | 0.10–1.70 | [51,52] |
| Irrigation system | MJ/ha | 373.7 | gCO$_2$eq | - | [53] |
| Use of diesel | MJ | - | gCO$_2$eq/MJ | 0.9 | [47] |
| **Transportation** | | | | | |
| Supplies | MJ/t*km | 0.87 | gCO$_2$eq/t.km | 71 | [47,54] |
| Biomass | MJ/t*km | 0.81 | gCO$_2$eq/t.km | 71 | [47,54] |

In order to elicit accurate and updated conclusions, the latest GHG lifetimes, radiative efficiencies and metrics are adopted, following the Sixth Assessment Report of the Intergovernmental Panel on Climate Change (IPCC) [58]. $CO_2$ equivalents in a time horizon of 100 years were calculated as follows: $CO_2 = 1$, $CH_4 = 27.9$ and $N_2O = 273$. The GWP values of the three chemical formulas were altered accordingly via the BioGrace-II GHG calculation tool. Nevertheless, the thorough analysis of emission coefficients is limited to specific inputs (agrochemicals and energy), whereas GHGs of other inputs (e.g., seeds, operations, etc.) are directly aggregated into $CO_2$ equivalents, based on the relevant remarks. The selection of the GWP100 indicator and the importance of the three major GHGs are also highlighted in other similar studies [25,59], taking into account short-term and mid-term implications of agricultural systems.

*2.5. Life Cycle Impact Assessment*

The stage where all the accounted emissions are transformed into environmental scores based on specific characterization factors is called Life Cycle Impact Assessment (LCIA) [60]. Apart from the mandatory process of calculating the environmental burden in the form of impact category indicators, other optional key elements (e.g., normalization, weighting, grouping, etc.) could facilitate the decision-making process and generate additional results [61]. The implications of scenarios reside not only in the differences among methodological choices, but in the time horizon of pollutants as well [60]. GWP is a common emission metric, which is calculated by the integration of radiative forcing over a specified time-period due to emission pulses [62]. GWP has been criticized as a forecast estimator related to temperature targets [63], though further improvements are still proposed in order to calculate emissions more effectively [64]. The conversion of climate pollutants into one unified indicator of $CO_2$ equivalents is based on the multiplication of the respective conversion factor with the quantity of each pollutant as follows [65]:

$$GWP = \sum Em \times Cf_{100} \qquad (1)$$

where *GWP* is the LCIA indicator in $CO_2$ equivalents and *Em* is the emissions of each pollutant. $Cf_{100}$ is the characterization factor for a time horizon of 100 years for each pollutant emitted by agricultural activities on the field. Nevertheless, agricultural production consists of manifold on-field activities and calculating GHGs via the emissions factor approach [66] in a more elaborate manner is necessary [25]:

$$CE_i = E_{agrochem}\left[\frac{kg\ CO_2}{ha}\right] + E_{en}\left[\frac{kg\ CO_2}{ha}\right] + E_{mach}\left[\frac{kg\ CO_2}{ha}\right] + E_{trans}\left[\frac{kg\ CO_2}{ha}\right] \qquad (2)$$

where $CE_i$ is the total kg of $CO_{eq}$ per agroforestry system. $E_{agrochem}$, $E_{en}$, $E_{mach}$, and $E_{trans}$ are the aggregated amounts of emissions from agrochemicals (fertilizers and pesticides, plus seeds) application, energy consumption (diesel, petrol, electricity, lubricants), machinery usage (operation, maintenance, manufacturing) and transportation of inputs and outputs to the respective destinations. Nitrogen fertilization is a parameter of direct and indirect emission augmentation, therefore, 1% of $N_2O$ emissions are considered as indirect [67].

Apart from the environmental assessment, the reciprocal influence of farm management to the economic and environmental sustainability of rural areas develops complex decision-making schemes, which is highlighted by many studies [67–69]. Therefore, the annual equivalent costs for the four production plans in Chalkidiki, following a Life Cycle Costing (LCC) approach are calculated as well. LCC takes into account the costs or cash flows, i.e., relative costs (both income and externalities, if included in the agreed scope) arising from acquisition and operation until the final disposal [70]. The full initial analysis of the LCC methodology and the parameters to be studied are presented in detail by Woodward [71]. The ultimate goal of LCC is to provide a framework for finding the overall cost of developing, producing, using and selling the product with the intention of

reducing overall costs [72] and in this context, a variety of LCC approach methods have been developed.

The Activity-Based Costing (ABC) method identifies costs through activities performed on cost targets (production or service activities), providing more accurate and traceable cost information [73]. The use of ABC can lead to the classification of value-added and non-value-added activities and allows the elimination of non-value-added activities. ABC is a life cycle assessment method with a similar scope to that known to ISO 14000 but has several significant advantages over the ISO approach [74]. ABC is relatively easier and provides a clear methodological framework, while its model is general and can be applied whenever activities are described in detail [75]. In agriculture, the approach consists of the decomposition of a project into factors (or activities) and the assignment of each production input cost to the respective factor, considering the actual level of consumption in each operation as follows:

$$TC_i = \sum C_{raw} + \sum C_{lab} + \sum C_{mach} + \sum C_{land} + \sum C_{en} + \sum C_{over} \tag{3}$$

where $TC_i$ represents the total costs for each agroforestry system $i$, $C_{raw}$ represents the aggregate costs for the acquisition of raw materials, $C_{lab}$ depicts the cost of labour, $C_{mach}$ is the capital service cost for machinery usage (including maintenance, insurance and the depreciation factor), $C_{land}$ represents the cost of landowning which depends on several factors, $C_{en}$ is the total costs for energy consumption (diesel, petrol and electricity) and $C_{over}$ accounts for the overheads for each system. Calculations and data analysis for the economic data are performed by the ABC software© v.2.1.2.0, which is an advanced cost analysis modelling software for the agribusiness industry (abcsoftware.org). Illustrations and graphs are created via the ABC software© v.2.1.2.0 and the R Studio Version 1.4.1106 [76].

## 3. Results

The first section of the results includes the depiction of emissions in $CO_2$ equivalents and the consumed energy in MJ for every cropping system in the area of Kassandra, based on the LCA approach. The economic and the LCA and LCC evaluation are also presented.

### 3.1. Emissions and Energy Consumed

The GHG emissions of every cropping system are divided into four separate sub-categories as listed in Table 2, in order to gain a clear perspective on the factors that significantly alter the emissions' total $CO_2$ equivalents. More specifically, the four sub-categories are (a) Agrochemicals and seeds, (b) Energy (electricity, diesel, petrol and lubricants), (c) Machinery (direct and indirect) and (d) Other (Transportation of inputs, indirect $N_2O$, etc.). The total GHG emissions of the two agroforestry systems (BOT and VBOT) are lower compared to the two monoculture cropping systems (BF and OT), while the agricultural practices for the OT reach a total amount of 2301.42 kg $CO_2$eq ha$^{-1}$. This figure (2301.42 kg $CO_2$eq ha$^{-1}$) is considerably higher than the rest of the cropping systems due to the requirements of OT mainly for energy and secondarily for agrochemicals, accounting together for more than 70% of the total emissions. Nevertheless, the emission pattern is quite similar among the cropping systems since the emissions for the Machinery and the Other sub-categories are significantly lower in comparison with the other two sub-categories (Agrochemicals and seeds and Energy). Regarding the VBOT system, the major difference hinges on the usage of 0-46-0 triple superphosphate fertilizer, achieving very low emission figures in the Agrochemicals and seeds (301.12 kg $CO_2$eq ha$^{-1}$) and Other (13.57 kg $CO_2$eq ha$^{-1}$) categories, due to the absence of nitrogenous chemical fertilizers.

Apart from the impacts to the environment, each activity in relation to farming practices imposes the consumption of energy. In Figure 2, a dual Y-axis bar chart is illustrated, depicting the total consumed energy in MJ ha$^{-1}$ and the total kg $CO_2$eq ha$^{-1}$ for each cropping system. In terms of energy consumption, the most demanding system is OT (22,968 MJ ha$^{-1}$), while the BOT system is the least energy intensive system (8547 MJ ha$^{-1}$). Nevertheless, practices adopted by BF and VBOT are also not considered energy demand-

ing (9027 MJ ha$^{-1}$ and 11,983 MJ ha$^{-1}$, respectively). Although the VBOT system emits lower levels of GHGs, the energy needed for all the relevant cultivation practices exceeds the ten-thousand marker by a significant margin (11,983.09 MJ ha$^{-1}$). This difference in the amount of energy consumed on the VBOT system is due to a mixture of fungicides and insecticides applied only on the olive trees of this system and not on the BOT system.

**Table 2.** GHG emissions for each cropping system in the region under study (kg $CO_2$eq ha$^{-1}$).

| Inputs | BF | OT | BOT | VBOT |
|---|---|---|---|---|
| Agrochemicals and seeds | 430.65 | 970.40 | 308.91 | 301.12 |
| Energy (electricity, diesel, petrol and lubricants) | 346.67 | 744.60 | 380.57 | 364.21 |
| Machinery (direct and indirect) | 66.47 | 149.48 | 72.70 | 129.28 |
| Other (Transportation of inputs, indirect $N_2O$, etc) | 210.30 | 436.94 | 155.69 | 13.57 |
| Total | 1054.08 | 2301.42 | 917.87 | 808.17 |

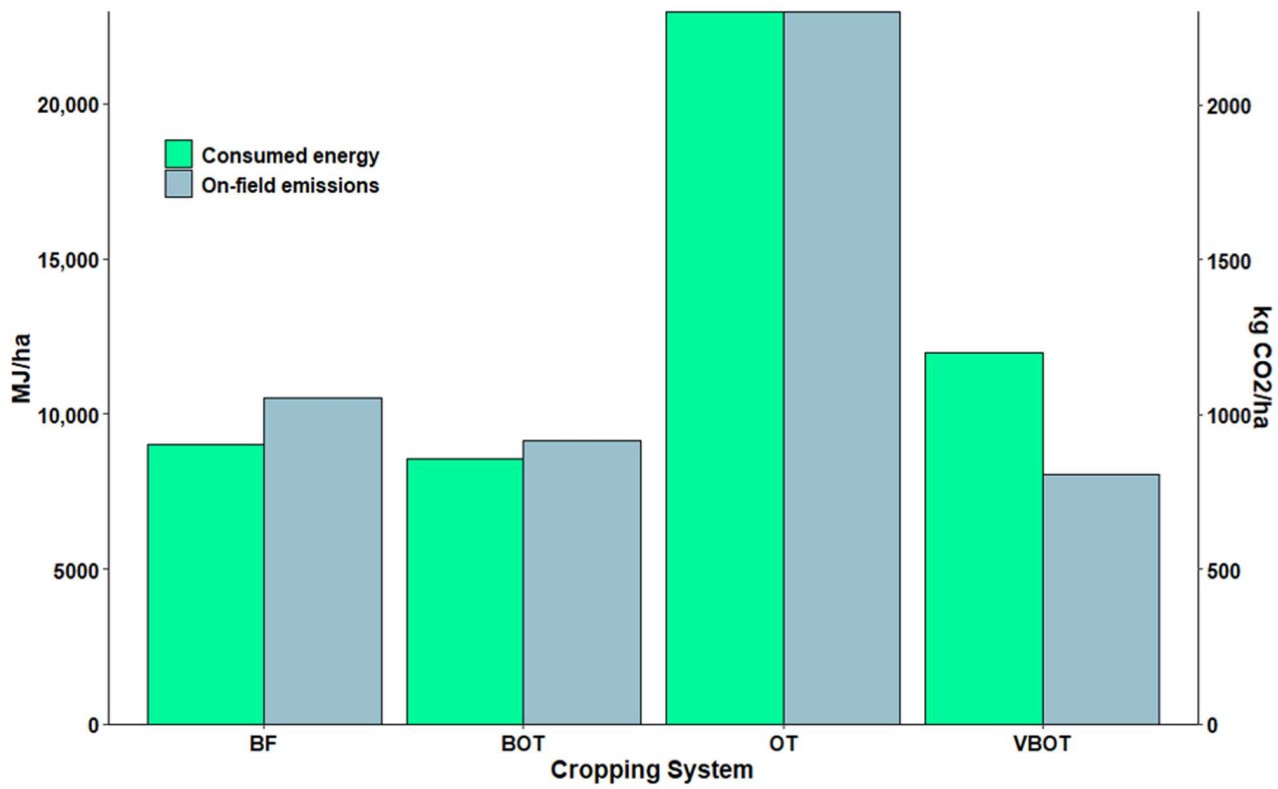

**Figure 2.** Consumed energy (MJ/ha) and on-field emissions for the four cropping systems.

### 3.2. Economic Outcome

The parallel goal of the current study is the economic assessment of the four cropping systems according to the LCC approach. In this context, essential economic data should be illustrated in order to present a clear perspective for the economic viability of each cropping system and how they are translated in economic terms. In Table 3, basic economic parameters are depicted, whilst the yield and gross revenue aspects for the two agroforestry systems (VBOT and BOT) generate a two-fold outcome. The economic assessment showed that the costs of BF cultivation exceed the sum total of gross revenue and subsidy generating a negative profit figure (EUR −53.44).

**Table 3.** Economic data for the four cropping systems.

| Cropping System | BF | OT | BOT | | VBOT | |
| | | | Barley | Olive Trees | Vetch-Barley | Olive Trees |
| --- | --- | --- | --- | --- | --- | --- |
| Annual yield (ton/ha) | 2.50 | 3.56 | 2.00 | 0.20 | 150 | 0.55 |
| Gross revenue (EUR/ha) | 325.00 | 3312.80 | 260.00 | 800.00 | 375.00 | 2216.00 |
| Total cost (EUR/ha) | 628.44 | 2805.08 | 486.94 | | 1776.49 | |
| Subsidy (EUR/ha) | 250.00 | 355.00 | 470.00 | | 680.00 | |
| Profit (EUR/ha) | −53.44 | 862.72 | 1043.06 | | 1494.51 | |

On the other hand, OT, BOT and VBOT are profitable cropping systems, justifying a total income per hectare of EUR 862.72, EUR 1043.06 and EUR 1494.51, respectively. The major difference among the cropping systems is the amount of total costs, which fluctuates from EUR 486.94 to EUR 2805.08. This is a significant margin, which is explained thoroughly via Figure 3 and Table 4. LCC is a methodological framework taking into account all the relevant costs of a product or process throughout its life cycle, while the Activity-Based Costing approach develops a segmentation of costs into specific activities, highlighting potential irregularities in the process. The two major aspects for the cost deviation in relation to OT and VBOT are land rent (44.3%) and labor (54.9%), respectively, while raw materials are to a greater or lesser extent about the same (Figure 3).

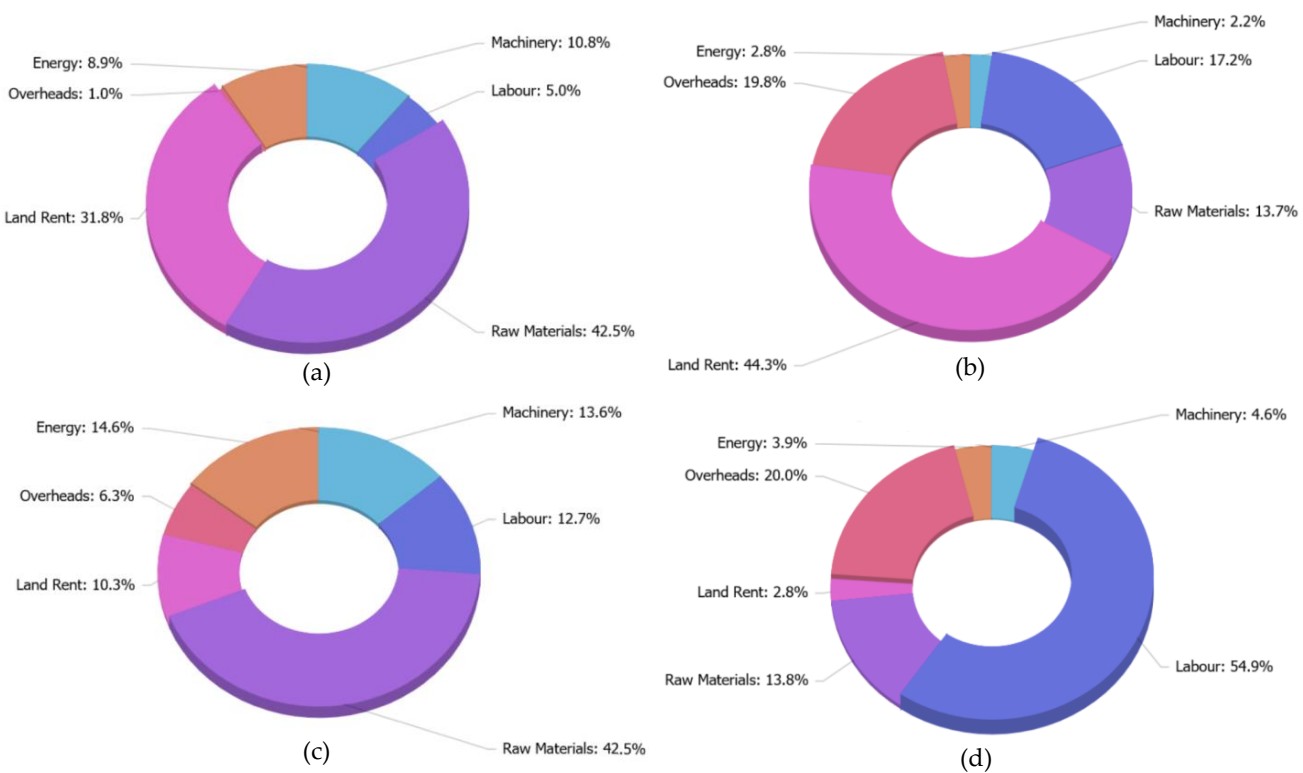

**Figure 3.** Shares of costs according to the LCC economic analysis for (**a**) Barley fodder (**b**) Olive trees, (**c**) Barley and olive trees and (**d**) Vetch–Barley and olive trees.

Focusing on other costs, the shares of energy are not significantly high with 14.6% for OT and 8.9% for BF, while the other two cropping systems are below 4% of the total costs. Moreover, overheads account for one-fifth of the total costs for OT and VBOT due to the remuneration of olive mills in kind, which translates into the cost figures. Comparing points of view, the economic analysis per activity illustrates important aspects which were difficult to identify otherwise (Table 4), neglecting land rent and focusing only on pure agricultural practices. For BF, sowing and fertilization are the two major activities costing more than 60% of the total expenses together. The benefit of agroforestry systems is illustrated via

the weed control activities which are performed only on the OT cropping system (EUR 346.53). Moreover, irrigation presents an extra cost for OT (EUR 41.23), since the other three systems are non-irrigated. Nevertheless, the interaction of agriculture and trees in BOT and VBOT integrates activities such as sowing and ploughing, which are not applicable to tree crops. Finally, the important cost factor of labor is explained for VBOT, as harvesting activities are performed only with human labor accounting for EUR 910.50.

**Table 4.** Economic analysis for each activity on the field.

| Activity | BF | OT | BOT | VBOT |
|---|---|---|---|---|
| Sowing | 140.55 | | 99.64 | 78.19 |
| Ploughing | 52.96 | | 42.02 | 36.18 |
| Fertilization | 113.55 | 240.43 | 85.25 | 64.41 |
| Herbiciding | 63.26 | 165.72 | 66.16 | 158.23 |
| Irrigation | | 41.23 | | |
| Weed control | | 346.53 | | |
| Prunning | | | 48.00 | 124.54 |
| Harvesting | 52.12 | 214.53 | 65.30 | 910.50 |
| Total | 422.44 | 1008.44 | 406.37 | 1372.05 |

The economic assessment though, illustrated that the different managerial approaches towards agroforestry could alter the perception of farmers in the region. More specifically, BOT and VBOT systems achieve considerable profit, though the total costs from activities differ to a significant degree, accounting EUR 406.37 and EUR 1372.05, respectively, and a gap of EUR 965.68.

*3.3. LCA and LCC Evaluation*

The significance of the gap between economic and environmental assessment could reveal externalities in relation to agricultural production [77], thus selecting the proper agricultural management approach is quite difficult. In this context, the results of LCA and LCC are illustrated via normalized accumulated waffle charts (Figure 4), representing costs (environmental and economic) in percentages. The waffle charts classify the orientation of economic and environmental impacts in percentage terms, depicting similarities and differences between the results.

More specifically, agrochemicals and seeds bear a detrimental environmental and economic consequence for every cropping system, except from the OT and the VBOT economic impacts, with the respective shares accounting for 24.57% and 14.17%, respectively. Furthermore, the environmental share of raw materials, regarding the LCA study, covers a significant amount of the normalized values. Energy requirements generate more than 30% of the total emissions, highlighting the importance of renewable energy sources. Nevertheless, the deviation between LCA and LCC results is more evident through the energy aspects, since the economic percentages are low in comparison to the respective ones of the environmental impacts. The huge gap between the two approaches hinges on human and machine labor, which creates a significant economic cost fluctuating between 23.22% and an extraordinary 61.30% of the total costs for VBOT. Therefore, the formulation of a cropping pattern including an assortment of agricultural activities, such as agroforestry systems, conceals numerous fluctuations between economic and environmental costs and should be integrated gradually to the production plans of farmers.

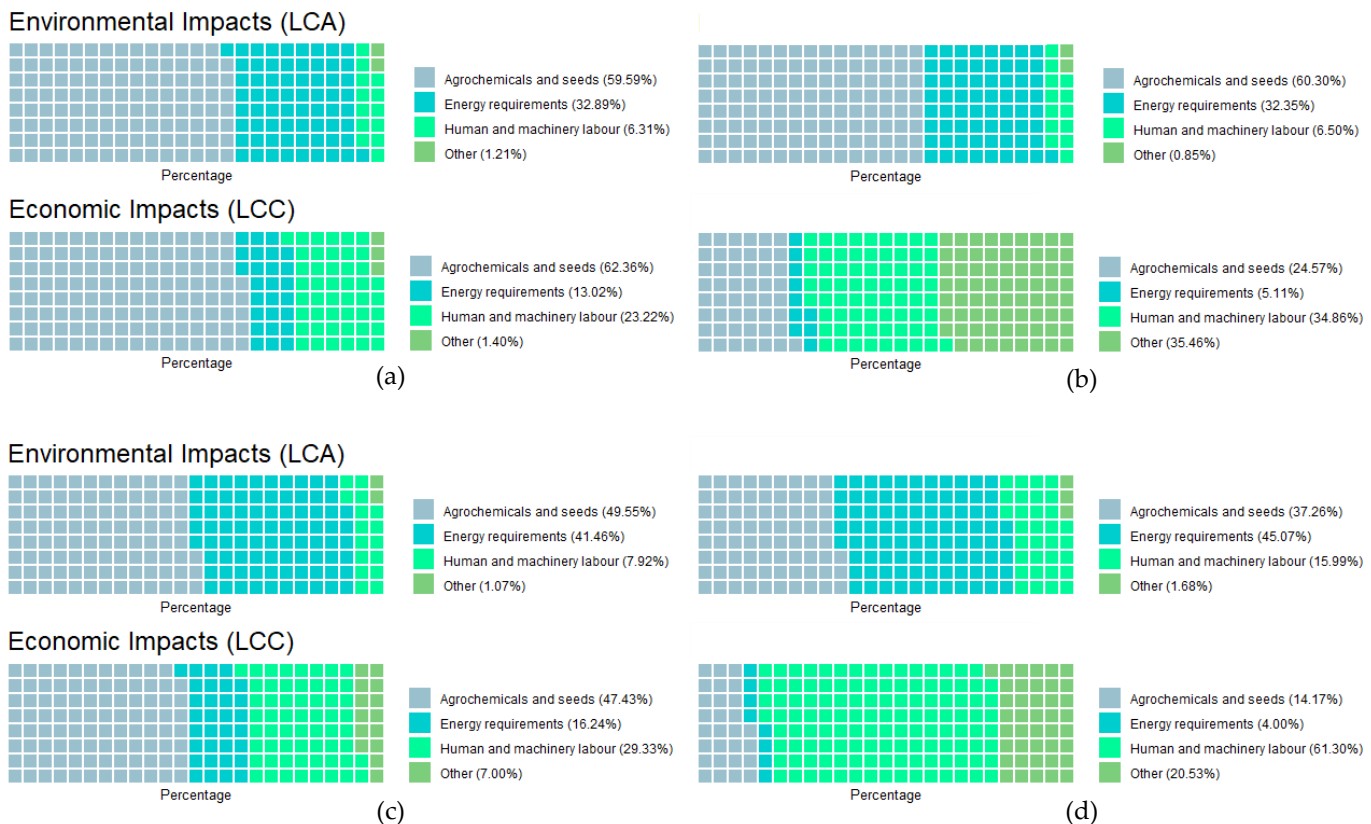

**Figure 4.** Waffle charts comparing the shares of impacts in LCA and LCC for: (**a**) Barley fodder (**b**) Olive trees, (**c**) Barley and olive trees and (**d**) Vetch–Barley and olive trees.

## 4. Discussion

The main focus of the study is the environmental and economic assessment of four cropping systems in Chalkidiki, in order to evaluate the sustainability factor of agroforestry in Greece. In this context, variation in the GWP can be observed in other agroforestry systems integrating olive trees. However, this variation is not significant, since silvopastoral and organic agroforestry systems in Italy emit 606 kg $CO_2$ eq. ha$^{-1}$ and 585 kg $CO_2$ eq. ha$^{-1}$, respectively [5]. These figures are close to those of the current study, though in the Italian study, synthetic fertilizers were not applied to the silvopastoral system and pesticides were not used for the organic system. On the other hand, findings of this study elicit the same results regarding the application and manufacture of fertilizers to olive production systems in Spain [78]. For all the investigated systems in Spain, fertilizers have contributed to the highest burdens in relation to environmental impacts.

Comparing the results to other studies integrating conventional agricultural practices in Greece, the GHG emissions from olive tree production accumulate to 2772.93 kg $CO_2$ eq. ha$^{-1}$ [25]. This figure is close to the respective environmental impacts of the OT system of the current study (2301.42 kg $CO_2$ eq. ha$^{-1}$), but far less than the two agroforestry systems (BOT and VBOT). Nevertheless, the balance between the environmental and economic scale is disrupted, since the economic gain from conventional agricultural practices (EUR 3686.60 ha$^{-1}$) is at least two-times greater than the ones of the current study. Similarly, De Luca et al. [79] highlight notable profitability rates for conventional agricultural practices in Italy, though the olive market price and the discount rate create significant fluctuations on the final profits.

The Activity-Based Costing approach presents a totally different perspective in olive tree production, since specific fixed costs are neglected on purpose focusing solely on the on-field practices. In this context, another study in Italy integrating olive trees in the production stage depicts annual operating costs equal to EUR 2032.0 ha$^{-1}$ and EUR 1667.7 ha$^{-1}$ for

organic and conventional agricultural practices, respectively. These figures are much higher than the BOT system of the current study; however, are close enough to the VBOT system considering that there is no irrigation cost for this agroforestry system. Another aspect to consider are the disparities in relation to labor compensations and rented services among the presented studies. For future work, the authors suggest the analysis of more midpoint and end-point indicators, in order to obtain a holistic view towards environmental sustainability of agroforestry systems in Greece. Furthermore, the assessment of more agroforestry systems could be integrated into a bio-economic model, taking into account future agricultural policies (CAP 2021–2027), environmental impacts (GWP indicator), climatic conditions (evapotranspiration, etc.) and farmers' profitability, generating an optimum production plan that mutually satisfies farmers and policy makers.

## 5. Conclusions

The assessment of olive farming systems, accounting for environmental and economic aspects of sustainability via an LCA and LCC approach was conducted by comparing four types of cropping systems that were applied in Kassandra. The environmental perspective of the study concluded that the agroforestry systems (BOT and VBOT) exhibited better results in comparison with monoculture farming due to the $N_2O$ emissions from N fertilizer application. The impact of N fertilization application is strongly connected to direct $N_2O$ emissions and the differences among the studied farming practices illustrated significant deviations of impacts due to intensive application of fertilizers in the OT system. Furthermore, the energy requirements for the OT cropping system (22,967.52 MJ ha$^{-1}$) were significantly higher than those of the other three cropping systems, while the VBOT system was the second most energy-demanding system with 11,983.09 MJ per hectare. Nevertheless, the concept of energy efficiency could provide a more thorough insight into the balance between consumed and potential generated energy from the selected cropping systems [80]. The energy-efficiency index could also highlight the effects of the selected cropping patterns and technologies on GHGs to the atmosphere [81].

Regarding the economic analysis, LCC generated interesting results in relation to the GWP indicator. More specifically, VBOT and BOT are less energy demanding, producing lower GHG emissions and simultaneously generating greater profits in relation to the monoculture farming systems in total. Nevertheless, if land rent is removed from the equation and based only on the activities performed on the field, VBOT is the most cost-intensive cropping system and BOT is the least. Therefore, the holistic assessment highlighted conflicting results regarding the shares of impacts in LCA and LCC. In economic terms, the cost of labor and other factors are more important when compared to the environmental impacts, in which agrochemicals and energy requirements play a more significant role. The field-based evidence from the current study can contribute to the facilitation of decision-making by policy makers, highlighting the LCA results and the respective environmental and economic impacts of agroforestry systems towards sustainable agricultural management.

**Author Contributions:** Formal analysis, conceptualization, data analysis, data curation, writing—review and editing, visualization, E.T.; Formal analysis, conceptualization, data analysis, writing—review and editing, S.I.; Formal analysis, conceptualization, data analysis, writing—review and editing, K.M.; Formal analysis, conceptualization, data analysis, D.K.; Formal analysis, data analysis, writing—review and editing, A.P. All authors have read and agreed to the published version of the manuscript.

**Funding:** This research received no external funding.

**Institutional Review Board Statement:** Not Applicable.

**Data Availability Statement:** Not Applicable.

**Conflicts of Interest:** The authors declare no conflict of interest.

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
