# Peer review of "Economic and Environmental Assessment of Olive Agroforestry Practices in Northern Greece"

_agriculture, doi:10.3390/agriculture12060851_

Round 1
Reviewer 1 Report
The article actually has a balance sheet and the totals are presented as bar charts. Little has been written about energy efficiency, for example. This is important in order to lay the foundations for research on energy efficiency. This shows how the emission of emission gases is balanced. I recommend that the authors read the article https://doi.org/10.3390/en14175574
There is no in-depth statistical analysis. Statistical analysis was not sufficiently discussed. The entire manuscript looks like an excerpt from a report from some major work. Without statistics and research, the article does not contribute to any knowledge development. An expansion of literature, conclusions and discussions is required.
Author Response
Point 1: The article actually has a balance sheet and the totals are presented as bar charts. Little has been written about energy efficiency, for example. This is important in order to lay the foundations for research on energy efficiency. This shows how the emission of emission gases is balanced. I recommend that the authors read the article https://doi.org/10.3390/en14175574
Response 1: Energy efficiency was not the main goal of this work. However, we will consider and take into account this comment for another upcoming research work. Citations of the relevant studies were added in the conclusions
Point 2: There is no in-depth statistical analysis. Statistical analysis was not sufficiently discussed.
Response 2: Statistical analysis was not included because it was included in the initial analysis framework
Point 3: The entire manuscript looks like an excerpt from a report from some major work. Without statistics and research, the article does not contribute to any knowledge development. An expansion of literature, conclusions and discussions is required.
Response 3: It is not. We are concerned that the reviewer does not identify the research objectives of this work. We will take his comment into account. Well noted, we took this comment into account.

Reviewer 2 Report
Overall, the paper addresses an interesting topic in a sensible manner. The comparison between two silvo-arable systems (one with a legume crop, one without) and two conventional cropping systems (specialized tree crop and arable crop) is a sensible one. The only lacking control is the ceral+legume arable system, but this is a minor flaw. Legume intercropping can be considered an addition to the cereal crop, in terms of sustainable crop nutrition and biodiversity, in line with the goals of an agroforestry system.
Nonetheless, some concerns need to be addressed:
- In my opinion, this is a system comparison trial, rather than a plot-scale agronomic trial. LCA is suited to investigate broader systems. You can upscale from the plot to the system, but the plots have to be rather homogeneous and representative at least of the field scale, and the variability should anyway be reported. This leads me to further questions
- How wide were the plots? Were they replicated?
- How were the yierly yield data used? If it´s a comparison between systems, and the plots are wide enough to be considered representative of a system, the year could provide reasonable variability, if spatial variability is not available and no extreme conditions were recorded in the given time interval
- related to this, no standard deviations were given in any chart/table of the study. This should be revised
- The number of trees ha-1 seems to be a crucial factor in this study. VBOT and BOT have 100 trees ha-1. What about the conventional OT? For sure trees are more dense here, but how much more? Is it a typical density? Was the age of the trees similar?
- What about straw use in BOT? If residues are left in the filed in VBOT but they are collected in BOT, a substantial difference between the systems has to be considered.
- Pages 4-6 have too much generic information on LCA/LCI/LCC and need revisions. For instance, L159-169 sounds quite repetitive. I suggest to focus this paragraph on the system boundaries rather than on the scope of the study
- VBOT and BOT: I think that the sum of the revenues from the arable crops and the olives should also be shown, including variability
- energy consumption in VBOT is remarkably higher than in BOT. Is it due to the different fertilization regime?
- related to this, can you show that P is not a limiting factor in BOT, BF and OT? The higher P fertilization rate in VBOT is justified by the leguminous crop, but soil analyses would help to clarify if P nutrition can be an issue in the other systems. In general, better information on the soil is recommended
- A breakdown of the different GHG gases is needed. A focus on N2O emissions is reported in the conclusions (L448) without sufficient discussion of this result, that is not shown in the charts and tables of the study, and this is not acceptable
- L455-457: is not the cost for land rent the same among systems? If yes, this sentence is not clear. If no, a major difference between systems is causing a counfounding effect, and land rent has to be excluded from the assessed costs, unless this difference is based on some intrinsic characteristic of the systems studied. If this is the case, this difference need to be properly explained. The more complex are the systems and the differences among them, the higher is the recommended study scale: a plot experiment may not be enough for complex assessments.
- A clearer, more informative abstract is needed. For instance, the four systems are not clearly indicated and some sentences need language revision (L14, L16, L29: some expressions do not sound enough scientific, e.g. "environmentally-friendly")
Minor issues are the following:
- L40: Agricultural intensification usually leads ro resource degradation, but not to increased land use, this is what you risk with extensification without compromises on yield. Please explain, provide references or rephrase
- L55, "integrate these adverse impacts" is not clear
- L72-L73. Are the 124.311 ha part of the 700.000 ha mentioned above? The relationship between the two areas is not clear
- L105, I suggest "environmental benefits" instead of "environmental friendly practices"
- L452-L461. The final part of the conclusions seems to be mixing environmental results with economic results in a not fully convincing and clear way. The last sentence is particularly unclear: how can economic factors be more important than environmental factors when it comes to money? This sounds obvious, maybe the meaning to be conveyed is different. It could be related to the economic cost of environmentally sound practices, or to the economic cost of agrochemicals and fertilizers, or else. It needs to be clarified.
Author Response
Please see the attachement

Reviewer 3 Report
The authors of this manuscript deals with the assessment of olive farming systems, accounting environmental and economic aspects of sustainability via an LCA and LCC approach was conducted by comparing four types of cropping systems that were applied in the Kassandra peninsula of Chalkidiki. The focus of the paper suits to a scope of the journal. The structure of the paper follows a structure of a scientific paper.
Abstract
The abstract is intended to provide the information needed to understand the context of the study, the objectives, the methods and the main results and conclusions. - I recommend reworking
Introduction
The introduction is written concisely and clearly - I have no comments.
Materials and Methods
The methodology is based on four production system in the Kassandra peninsula. Based on this, the research methodology was determined, where the gist is the evaluation of trade-offs between the economic and environmental sustainability of Agroforestry systems, in comparison to conventional agricultural practices, which are dominant in the area. The methodology of the studies is described clearly, concisely and determined correctly. I have no comments on the experiment methodology.
Results
The form of evaluation of the results "The less energy consuming system is BOT followed closely by BF, .." is unscientific. I recommend reworking the verbal evaluation of the results. Graphs and tables correspond to the classical evaluation of scientific results.
Discussion
The discussion is a critical comparison of the results with the works of other authors - I have no comments.
Conclusion
The conclusions need to be revised to emphasize the contribution of the work and the results achieved to scientific knowledge. Indicate the limitations of the research provided and the direction of future research. The analysis was performed in the period 2014-2016 (eight years ago). The research raises the question, what are the current results? I recommend the authors to supplement the conclusion with current information in the given area of Kassandra. This will increase science and relevance and will have a high scientific added value.
Author Response
Response to Reviewer 3 Comments
Abstract
The abstract is intended to provide the information needed to understand the context of the study, the objectives, the methods and the main results and conclusions. - I recommend reworking
Thank you for your comment, a more informative abstract is included
Introduction
The introduction is written concisely and clearly - I have no comments.
Thank you for your comment
Materials and Methods
The methodology is based on four production system in the Kassandra peninsula. Based on this, the research methodology was determined, where the gist is the evaluation of trade-offs between the economic and environmental sustainability of Agroforestry systems, in comparison to conventional agricultural practices, which are dominant in the area. The methodology of the studies is described clearly, concisely and determined correctly. I have no comments on the experiment methodology.
Thank you for your comment
Results
The form of evaluation of the results "The less energy consuming system is BOT followed closely by BF, .." is unscientific. I recommend reworking the verbal evaluation of the results. Graphs and tables correspond to the classical evaluation of scientific results.
ok, rephrased
Discussion
The discussion is a critical comparison of the results with the works of other authors - I have no comments
Conclusion
The conclusions need to be revised to emphasize the contribution of the work and the results achieved to scientific knowledge. Indicate the limitations of the research provided and the direction of future research. The analysis was performed in the period 2014-2016 (eight years ago). The research raises the question, what are the current results? I recommend the authors to supplement the conclusion with current information in the given area of Kassandra. This will increase science and relevance and will have a high scientific added value.
Thank you for your comments. The conclusions are enriched with added information

Round 2
Reviewer 1 Report
The article was well-prepared in accordance with the comments.
Author Response
Thanks for the comments
Reviewer 2 Report
To be honest, the authors did not provide acceptable answers to many of my remarks (especially those in the first page of the "author response" file). For instance, they don´t provide the required information about tree density and replication. They say there is spatial replication, then they contradict themselves saying that the three study years were considered as replicates. All this considered, no variability is shown in the study, so we don´t know what was the replication for. They say they shown the breakdown of the emission into different gases, but I couldn´t find it in the study. Some improvements are visible, but the language still needs improvement. Overall, I am sorry to say that it seems to me that their revisions and their answers were really hasty.
Author Response
Point 1. To be honest, the authors did not provide acceptable answers to many of my remarks (especially those in the first page of the "author response" file). For instance, they don´t provide the required information about tree density and replication. They say there is spatial replication, then they contradict themselves saying that the three study years were considered as replicates. All this considered, no variability is shown in the study, so we don´t know what was the replication for.
The tree density of OT is 250 trees/ha. The age of trees is 40 years old and they are managed (pruning) like VBOT and BOT production systems (a phrase added in the description of production systems).
Regarding the replications there were three replications and the design was a latin square. More information you can find in the following paper. The 3 years are not the replications. We measure for 3 years in order to avoid climate variations.
Intercrop of olive trees with cereals and legumes in Chalkidiki, Northern Greece. Mantzanas et. al. 2021 Agroforest Syst 95, 895–905 DOI: 10.1007/s10457-021-00618-6
Point 2. They say they shown the breakdown of the emission into different gases, but I couldn´t find it in the study. Some improvements are visible, but the language still needs improvement. Overall, I am sorry to say that it seems to me that their revisions and their answers were really hasty.
A phrase was added in order to clear confusion about GHG breakdown. GHGs which were highlighted by the BIOGRACE standard values (relevant excel tool, available from: https://www.biograce.net/biograce2/content/ghgcalculationtool_electricityheatingcooling/overview ) provide information for all the chemical formulas (CO2, CH4, N2O) and the respective emission coefficients, while the other remarks provide only the unified CO2 eq value. Therefore, we did not analyze the GHG inventory (Table 1) because it could be confusing for the reader and we have provided only the CO2 eq values.
